# Polymeric Microneedles for Transdermal Delivery of Rivastigmine: Design and Application in Skin Mimetic Model

**DOI:** 10.3390/pharmaceutics14040752

**Published:** 2022-03-30

**Authors:** Tânia M. T. Guimarães, Tânia Moniz, Cláudia Nunes, Maya Margaritova Zaharieva, Mila Kaleva, Krassimira Yoncheva, Hristo Najdenski, Sofia A. Costa Lima, Salette Reis

**Affiliations:** 1LAQV, REQUIMTE, Department of Chemical Sciences, Faculty of Pharmacy, University of Porto, Rua de Jorge Viterbo Ferreira, 228, 4050-313 Porto, Portugal; maritani98@hotmail.com (T.M.T.G.); tmoniz@ff.up.pt (T.M.); cdnunes@ff.up.pt (C.N.); shreis@ff.up.pt (S.R.); 2Department of Infectious Microbiology, The Stephan Angeloff Institute of Microbiology, Bulgarian Academy of Sciences, 26 Acad. G. Bonchev Str., 1113 Sofia, Bulgaria; zaharieva26@yahoo.com (M.M.Z.); milakalevavet@abv.bg (M.K.); hnajdenski@gmail.com (H.N.); 3Department of Pharmaceutical Technology, Faculty of Pharmacy, Medical University of Sofia, Dunav Str. 2, 1000 Sofia, Bulgaria; krassi.yoncheva@gmail.com

**Keywords:** alginate, calcein, carrageenan, cutaneous administration, dementia, hydrogels, transdermal patch

## Abstract

In the last years, microneedles (MNs) have been considered a valuable, painless, and minimally invasive approach for controlled transdermal drug delivery (TDD). Rivastigmine (RV), a drug administered to patients suffering from dementia, is currently delivered by oral or transdermal routes; however, both present limitations, mainly gastrointestinal adverse symptoms or local skin irritation and drug losses, respectively, for each route. Given this, the objective of the present work was to develop and evaluate the potential of polymeric MNs for RV transdermal delivery in a controlled manner. Polymeric MNs with two needle heights and different compositions were developed with calcein as a fluorescent model molecule. Morphology and mechanical characterisation were accessed. Skin permeation experiments showed the ability of the devices to deliver calcein and confirmed that the arrays were able to efficiently pierce the skin. To obtain a new TDD anti-dementia therapeutic solution, RV was loaded in 800 µm polymeric MNs of alginate and alginate/k-carrageenan MNs. In the presence of RV, the MN’s morphology was maintained; however, the presence of RV influenced the compression force. Skin permeation studies revealed that RV-loaded MNs allowed a more efficient controlled release of the drug than the commercial patch. In vivo, skin irritation tests in rabbits revealed that the developed MNs were innocuous upon removal, in contrast with the evidence found for Exelon^®^, the commercial patch, which caused slight mechanical damage to the skin. The herein-produced MNs demonstrated a more controlled release of the drug, being the more suitable option for the transdermal delivery of RV.

## 1. Introduction

Dementia is a clinical syndrome that is characterised by the deterioration of cognitive functions [1] and is one of the most common symptoms in Alzheimer’s (AD) and Parkinson’s (PD) diseases [2]. Current pharmacological options for neurodegenerative dementia aim to restore the disrupted neurotransmission caused by cholinergic neuron damage. Acetylcholinesterase inhibitors (AchEI) and *N*-methyl-D-aspartic acid (NMDA) receptor antagonists are two classes of drugs currently approved for the treatment of dementia. Within the first group lies donepezil, rivastigmine (RV), and galantamine, while in the second one, memantine is one of the most well-known antagonists [3] (Figure 1).

RV presents an advantage in comparison to other inhibitors available in the market since it inhibits both cholinergic enzymes, AchE and butyrylcholinesterase. RV is formulated as hard capsules or oral solutions, both of which are approved for the symptomatic control of mild to moderate dementia triggered by AD and PD [4]. However, these forms of RV are associated with undesired side effects, namely gastrointestinal adverse symptoms and rapid maximum plasma concentration that hamper the optimal therapeutic dose to be achieved [5]. Therapeutic treatment of dementia is also available as transdermal patches for RV release (Exelon^®^) (4.6 mg [5 cm^2^], 9.5 mg [10 cm^2^], and 13.3 mg [15 cm^2^]) [6]. Exelon^®^ patches are a single-use, prescription-only, 24 h release patch approved in the United States of America (USA) [7] and in Europe [8]. The use of an RV patch improves patient compliance and avoids overdosage or non-administration of the medication, especially in the elderly population [9]. However, the use of the patches requires a greater amount of drug in the device to obtain the intended plasmatic drug concentration (4.6–13.3 mg released/24 h) [10]. In addition, reports show rapid drug release of RV [11] and problems associated with local skin irritation [12]. Given this panorama, the development of novel therapeutic tools for skin application is of the utmost importance.

Novel strategies have been described applying different RV delivery systems, especially for intranasal delivery [13]. Nanoparticles (NPs), particularly polymeric NPs, have been extensively studied to deliver RV to the brain [14]. As an example, Fazil and co-workers developed RV-loaded chitosan NPs, with enhanced bioavailability and uptake of RV through the brain [15]. Other types of NPs, namely solid lipid nanoparticles comprising RV, have also been developed with the same purpose [16]. Nanoformulations were also designed for alternative administration routes, namely parenteral [17] and intravenous [18,19]. Many other NPs have been designed to improve the delivery of RV [11,20] by considering transdermal administration. Recently, an innovative strategy using a transdermal patch as a base was proposed by Cai and colleagues [21]. Sonophoresis application has also been reported [22].

As one of the most promising approaches for the improvement of transdermal drug delivery (TDD), microneedles (MNs) are a minimally invasive mechanical approach for the enhancement of drug permeation while overcoming some of the limitations presented by conventional systems [23,24]. Given that, a larger number of pharmaceuticals and cosmetics can be delivered using MNs, expanding the number of drugs suitable for transdermal delivery [25,26]. A hollow MN system was developed for the delivery of RV in liquid format through the skin [27]. The work describes a 3M round-shaped device that is used for the delivery of RV, and then the patch is immediately removed. Nevertheless, the authors only reported the development of the device and the evaluation of the solution stability over time. As far as we know, this is a unique report describing a non-polymeric MN system for the delivery of RV.

Many biocompatible, biodegradable, and water-soluble polymers have been studied as drug delivery systems, given their high loading capacity of drugs within the polymeric matrix. Drug loading within the polymeric MN facilitates the controlled release of a given biopharmaceutical product, the prospect of reaching specific tissues, and the ability to respond according to different stimuli [28]. Furthermore, granted the viscoelastic properties presented by polymers, mechanical strength related to shear breakage is enhanced [29]. Particularly, marine polysaccharides obtained from algae, animals, and microorganisms represent a sustainable, cheap, and non-toxic source of polymers that preserve the beneficial features of biodegradability and biocompatibility like the remaining polymers. Additionally, these polysaccharides can exhibit anti-inflammatory and antimicrobial properties [30,31]. The most pertinent marine-origin polysaccharides for this study are alginate and k-carrageenan (k-CRG), although many other polymers like chitosan and hyaluronic acid manifest relevance within the scientific community as drug delivery systems [25]. Despite the numerous alginate-derived MNs described in the literature, to the best of our knowledge, there is no record of MNs made from k-CRG per se or combined with alginate.

The goal of this study was the development of an MN array capable of systemic delivery of RV through the skin in a controlled manner able to improve dementia therapeutic options currently available. The study assessed the use of two hydrogel-based MNs made from natural polymers, alginate, and k-CRG using two types of MN moulds (400 vs. 800 µm in height). After the production and optimisation of the MNs’ fabrication process, the characterisation involved morphological analysis, evaluation of their compressive strength in comparison with drug-free MNs employing a texture analyser and in vitro porcine skin permeation assays. Alginate and alginate/k-CRG hydrogels containing calcein, a model drug, were applied in the optimisation process. Calcein was employed to optimise MN production and later applied to RV-loaded devices and to characterise the prepared devices, taking advantage of its fluorescent properties. Then, RV-loaded MNs were formulated and characterised as defined above. The present work will certainly contribute to the development of new TDD systems for RV, aiming to overcome drawbacks associated with currently available transdermal strategies.

## 2. Materials and Methods

### 2.1. Materials and Instrumentation

Sodium alginate was purchased from ACROS Organics^TM^ (Thermo Fisher Scientific, Massachusetts, MA, USA). Calcein, k-CRG, RV tartrate, and Dulbecco’s phosphate-buffered saline (PBS) (10×) were acquired from Sigma-Aldrich (St. Louis, MO, USA). Exelon^®^ patches were acquired from Novartis. Acetonitrile was acquired from Fisher Scientific (Hampton, NY, USA). Silicone moulds were obtained from Micropoint Technologies (Pioneer junction, Singapore). Fresh porcine ears obtained from different adult animals were purchased in a local slaughter (Porto, Portugal). Double-deionised water was provided by an ultra-pure water system (Arium Pro, Sartorius AG, Gottingen, Germany). The reagents were weighted in a digital analytical balance Kern ACJ/ACS 80-4 (Kern & Sohn; Balingen, Germany). pH measurements were achieved using a JENWAY 550 pH meter (Staffordshire, UK). Texture analysis was performed using a TA.XT2 Texture Analyser (Stable Micro Systems Ltd., Haslemere, UK). SEM and confocal fluorescence microscopy studies were performed at the Laboratory for Scanning Electron Microscopy and X-Ray Microanalysis and High-Resolution Microscopy Unit, respectively, both at CEMUP (Materials Centre of the University of Porto, “Centro de Estudos de Materiais da Universidade do Porto”). Histological samples have been processed in the histological service of “ICBAS, Universidade do Porto”.

### 2.2. Hydrogel Preparation

#### 2.2.1. Preparation of the Alginate and k-Carrageenan Hydrogels

Alginate and k-CRG solutions were prepared by dissolving the alginic acid and k-CRG powders in double deionised water at a final concentration of 7% and 1% (*w*/*w*), respectively. Solutions were placed in a magnetic stirring plate (IKA-Werke, Staufen, Germany) at 60 °C, under continuous stirring, until a homogeneous solution was obtained. After that, the polymers were allowed to cool down before preparing the mixture of alginate and k-CRG, obtained by weighing and physically mixing the two polymeric solutions in 1:1 (*w*/*w*) proportion.

#### 2.2.2. Preparation of the Drug-Loaded Hydrogels

Firstly, calcein-loaded hydrogels were prepared for further production of calcein-loaded MNs, which were useful for the optimisation of the production method and the visualisation of the permeation of the model compound by using its fluorescent properties in confocal microscopy analysis. Calcein, at a concentration of 0.1% (*w*/*w*), was initially added (as powder) to each hydrogel formulation, mixed with a spatula, and then placed into a sonication bath (Sonica^®^ Ultrasonic Cleaner, Shenzhen, China) for a few minutes to ensure all calcein molecules were incorporated into the hydrogel. Once the three mixtures (alginate, k-CRG, and alginate/k-CRG) were prepared, aluminium foil was used to protect the samples from the light.

RV-loaded hydrogels were produced in the same manner, taking into account the mass of RV per cm^2^ of Exelon^®^ patch. Assuming a uniform distribution of RV in the patch, Exelon^®^ Patch 5 cm^2^ presents 1.8 mg of drug per cm^2^. The microneedles present 0.5 cm^2^ of area, which translates into the addition of 0.9 mg of RV per MN array in this study (0.9 mg/150 mg hydrogel).

### 2.3. Preparation and Optimisation of the Microneedles’ Fabrication Process

Microneedles were produced using two types of silicon moulds, differing on the height and base of the needle. All MN moulds had a total of 100 needles, which translated into an array of 10 × 10. MNs pyramid base has 150 or 200 µm and a height of 400 or 800 µm, respectively.

The first trials included the preparation of devices considering the three previously described hydrogels (alginate, k-CRG, and alginate/k-CRG), adapting a previously described procedure [32]. The experimental conditions defined as Method 1 are summarised in Table 1. Since this methodology was inefficient given the loss of polymer that was observed after each centrifugation cycle (please see Section 3), another three protocols were developed (Table 1). The fourth method was chosen to produce the devices. Briefly, 25 mg of each hydrogel was loaded into the respective MN moulds and then placed in a 12 well plate. Following this, the MN moulds were centrifuged (Eppendorf centrifuge, 5810R, Madrid, Spain) at 3500 rpm and 30 °C for 1 h. The second layer of 25 mg of hydrogel was loaded, and the centrifugation step was repeated under the same conditions. Then, the moulds were kept overnight in a climate room at 21 °C, in a Petri dish sealed with parafilm and protected from light. On the second day of MN production, the third layer of 50 mg of the formulation was added to the moulds, followed by centrifugation, as mentioned above. This step was repeated on the third day after overnight incubation under the same conditions. For the drying process, filled moulds were placed in an incubator shaker (ES-60 Incubator Shaker, Miulab, Hangzhou, China) at 100 rpm, at 25 °C with fan for 3 h or until the corners of the MNs appear to be detaching from the MN mould. Empty MNs (without any compound) of each hydrogel (alginate, k-CRG, or mixture) were prepared as the control for morphological and texture analysis.

### 2.4. Characterisation of the Microneedle Devices

#### 2.4.1. Topography Analysis—Scanning Electron Microscopy

Surface morphology of fabricated MNs was visualised by SEM using an FEI Quanta 400 FEG ESEM/EDAX Pegasus X4M (Zeiss, Jena, Germany) with an accelerating voltage of 10 kV for high-resolution imaging. MN arrays were mounted onto metal pins with double-sided carbon tape and coated with an ultra-thin layer of gold/palladium (Au/Pd) to increase surface conductivity. Drug-free MNs, calcein-loaded, and RV-loaded MNs were observed using a magnification of 75× and 250× to get a general image of the patch and a more detailed look at the needle structure, respectively. All samples developed were analysed in duplicates.

#### 2.4.2. Texture Analysis

To determine the compression force for each type of MN, mechanical characterisation was performed using a TA.XT2 Texture Analyser (Stable Micro Systems Ltd., Haslemere, UK).

MNs were placed on a flat rigid surface of a stainless steel base plate, and an axial compression load was applied through a metal probe of 2 mm of diameter moving towards the baseplate with a 0.3 mm path length, at 0.01 mm/s. For each MN device, at least 5 readings were made, one in the central part of the MN array and one at each corner. At least 3 independent devices were analysed. Considering probe diameter and design and dimensions of the MN array based on manufacture details, it was estimated that the force measured by the probe may correspond to approximately 2 MNs per reading. Thus, the values of forces obtained were divided in half to get insight into the fracture force per needle.

### 2.5. Skin Permeation Studies

Fresh porcine ear skin was obtained at a local slaughterhouse, preferably from healthy unmarked animals, given that compromised skin can lead to false results [33]. Five porcine ears were acquired from female pigs with ages between 8 and 14 months. Initially, the skin was inspected to locate any possible lesion sites or visible capillary veins that would need to be dismissed for future studies. Next, the external part of the pig ear was separated from the underlying cartilage with the help of a scalpel, and any subcutaneous fat tissue was removed to better isolate epidermal/dermal tissue. Samples obtained were wrapped in aluminum foil and stored at −20 °C until required, for a maximum of one month. After permeation studies, the skin was immediately processed without any freezing process before formaldehyde fixation or skin digestion for RV quantification.

For permeation studies, full-thickness skin was allowed to thaw at room temperature for approximately 30 min before being cut into circles with a diameter equivalent to the donor compartment (approximately 2 cm). Whole skin permeation studies for both the model drug (calcein) and RV were performed using Franz Diffusion Cells (9 mm unjacketed Franz Diffusion Cell with 5 mL receptor, O-ring joint, clear glass, clamp, and stir-bar; PermeGear, Inc., Hellertown, PA, USA). As a control for the study, empty MNs were also used. The receptor chamber was filled with 4.7 mL of PBS buffer pH 7.4, and a magnetic bar was added, allowing the system to reach 37 °C on the magnetic stirring plate. MNs were placed in the centre of the skin circles and secured with medical adhesive tape. Then, the pressure was applied with the index finger for 10 s, followed by the application of a 1.6 N force using the applicator (Micropoint Technologies, Pioneer junction, Singapore) for the same period of time. Both calcein solution in PBS (0.15 mg of drug per 150 µL PBS) and hydrogels containing the drugs (calcein or RV, 0.15 mg or 0.9 mg per 150 mg of hydrogel, respectively) were used as controls. Exelon^®^ patches (0.5 cm^2^ from an Exelon^®^ patch 5 cm^2^) were also used as a control for the RV-loaded MN study and applied on the skin according to the instructions.

The system was mounted with the *stratum corneum* (*SC*) facing upwards, between the donor and receptor compartments, with a diffusion area of 0.64 cm^2^, and held in place with a metal clamp. To prevent evaporation, the donor chamber opening was sealed with Parafilm^®^. Aliquots of 200 µL were collected from the receptor sampling port at set times of 3, 6, 8, and 24 h, and the same amount of fresh preheated PBS medium was reintroduced into the receptor. All the conditions were carried out, at least, in triplicate (*n* = 3) under continuous stirring at 37 °C for 24 h and protected from light exposure to prevent possible degradation of the compounds. The skin used in the permeation studies for each condition, as well as the contents of the donor chamber, were collected and stored at −20 °C until further analysis.

#### 2.5.1. Quantification of Drug Permeation

A standard curve was prepared for calcein from a stock solution obtained by dissolving the model compound in PBS buffer pH 7.4 with the help of a sonication bath (Sonica^®^ Ultrasonic Cleaner, Merseyside, England), at a final concentration of 0.622 mg/L. The standards were obtained by dilution from the stock at concentrations of (7.8, 3.89, 1.95, and 0.97) × 10^−3^ mg/mL in PBS. Fluorescence intensity of the solutions was measured in a microplate reader (Synergy™ HT Multi-Detection Microplate Reader, Cytation^®^, Santa Clara, CA, USA) with excitation and emission wavelengths of 485 nm and 528 nm, respectively. At least three independent curves were analysed, thus resulting in an average of: fluorescence intensity (a.u.) = 3.63 × 10^6^ [calcein, mg/mL] + 2.63 × 10^3^, with R^2^ = 0.990.

Using the calibration curve, calcein concentration was calculated considering the fluorescence in each aliquot retrieved from the permeation study, as previously described [34,35]. Mass of calcein per 4.7 mL of PBS was obtained using Equation (1):(1)$$C_{c}=\frac{m_{c}}{V_{t}}$$
where $C_{c}$ represents calcein concentration in the acceptor chamber (mg/mL), $m_{c}$ the mass of calcein (mg), and $V_{t}$ the PBS volume in the acceptor chamber (4.7 mL).

The obtained values were used to mathematically evaluate the apparent permeability coefficient ($P_{app}$) and the % of permeation (% *P*) using the formulas, respectively:(2)$$P_{app}\left(cm/s\right)=\frac{\sum m_{a}}{A\cdot m_{0}\cdot t}$$
(3)$$\%\;P=\frac{\sum m_{a}}{m_{0}}\;\times \;100\;$$
where $\sum m_{a}$ represents the sum of the drug mass permeating over time in the acceptor chamber (g), A is the area of diffusion in the Franz Diffusion Cell system (0.64 cm^2^), $m_{0}$ is the initial donor mass (g), and t the time (seconds).

For RV quantification, high-performance liquid chromatography (HPLC) analysis was conducted. Analysis was performed using a reversed-phase monolithic column (Chromolith^®^RP-18e,100 mm ×4.6 mm i.d., Merck), connected to a Jasco (Easton, PA, USA) HPLC system (pump PU-4180, autosampler AS-4050 and LC-Net II/ADC controller) coupled to a PDA detector (JascoMD-4010, Start Wavelength = 200 nm, End Wavelength = 400 nm). Data processing was performed by ChromNAV2.0 HPLC software (JASCO, Easton, PA, USA). The UV chromatogram was acquired at 214 nm with a retention time of 4.5 min. The mobile phase consisted of PBS and acetonitrile (70:30, *v*/*v*) with pH adjusted with HCl to 4. The flow rate of the system was 1 mL/min. All samples investigated had the pH adjusted to 4 (JENWAY 550 pH meter, Staffordshire, UK).

To establish the calibration curve, a stock solution of 1 mg/mL of RV in PBS 7.4 was diluted in the mobile phase to obtain standard concentrations ranging from 0.2 to 0.01 mg/mL (0.2, 0.1, 0.05, 0.02, 0.01 mg/mL). At least three independent curves were recorded thus resulting in an average of: area under the curve (AUC) = 1.99 × 10^7^ [RV, in mg/mL] − 2.75 × 10^4^, with R^2^ = 0.999. The same calculation method mentioned above for the model drug was used for RV (Equations (2) and (3)).

#### 2.5.2. Quantification of Drug Remaining in the Apical Compartment

The amount of sample remaining in the apical compartment was carefully removed at the end of the permeation studies (after 24 h). Then, 30 mg of each condition in the study was dissolved in 200 µL of PBS pH 7.4 or in 350 µL of HPLC mobile phase, respectively for calcein or RV. Next, samples were vortexed for 2 min and then left for 30 min in a sonication bath followed by centrifugation at 10,000 rpm at 20 °C for, at least, 15 min (Allegra X-15R, Beckman Coulter, CA, USA). The visible pellet formation was discarded, and the supernatant was recovered and analysed by fluorescence spectroscopy or HPLC, respectively, for calcein or RV, as described above. All samples are resultant from the independent assays described in Section 2.5.

#### 2.5.3. Quantification of the Drug Retained in the Skin

After disassembling the Franz Diffusion Cell system, porcine skin was gently washed. Regarding calcein, instead of skin digestion for quantification of the retained drug, the mass of the compound was estimated given the mass of calcein quantified in the apical and basolateral compartments, using the following equation:(4)$$\%\;Retained\;drug=100-\left(\;\%\;\mathrm{of}\;remaining\;drug\;in\;the\;apical-\;\%\;P\right)$$
where % of the remaining drug in the apical represents the percentage of calcein in the apical after 24 h and the percentage of the permeated drug (% *P*) into the basolateral at 24 h.

To determine the mass of RV retained in the skin, each tissue sample was placed in a 50 mL centrifuge tube, and approximately 5 mL of mobile phase was added before ultraturrax digestion (Ultra-turrax T25, Janke & Kunkel IKA-Labortechnick, Staufen, Germany) at 12,000 rpm for about 2 min, or until the skin sample was destroyed. Each sample was then placed in a sonication bath for 30 min and immediately centrifuged at 4750 rpm for 20 min (Allegra X-15R, Beckman Coulter, CA, USA). The supernatant was collected and left to evaporate in a rotary evaporator (Rotary evaporator, Buchi, R200, Flawil, Switzerland) using a stream of argon and a thermostatic bath at 40 °C until the sample was fully dry. Samples were then resuspended in 500 µL of mobile phase and centrifuged at 10,000 rpm for 15 min. HPLC analysis was carried out as previously described. All samples are resultant from the independent assays described in Section 2.5.

#### 2.5.4. Histology of the Skin

Skin samples resulting from calcein permeation assays were fixed with formaldehyde and processed. Histological sections were obtained from skin resultant from two independent permeation assays. Briefly, tissue was paraffin-embedded and sectioned at a thickness of 5 μm. Then, hematoxylin and eosin (H&E) stained samples were analysed by optical microscopy (Inverted Optical Microscope, CK2 model, Olympus, Hamburg, Germany), equipped with a digital camera (MotiCam S12) and using 4× or 10× objectives. To further investigate pore-forming structures in the skin, unstained samples were analysed using confocal fluorescence microscopy (Leica Stellaris 8 confocal microscope, Leica Microsystems, Wetzlar, Germany) equipped with the Leica Application Suite X package (LAS X). Images were acquired, following excitation at 495 nm by a HyD X detector (510–600 nm), with a resolution of 1024 × 1024 using a 10X/0.4 objective, and then processed under ImageJ software (Fiji app, ImageJ2 version, freely available at http://imagej.net/, accessed on 27 January 2022).

### 2.6. Skin Irritation Test

The potential of alginate and alginate/k-CRG MNs to produce dermal irritation was assessed according to the standard protocol in ISO 10993-10 [36]. Experimental rabbits were grown in the animal house of the Stephan Angeloff Institute of Microbiology (Reg. Nr. 1113-0005). The Certificate to work with rabbits (Nr. 232/2020) was issued by the Bulgarian Food Safety Agency at the Ministry of Agriculture and Food based on approval by the National Commission of Animal Ethics. Briefly, six healthy young New Zealand white rabbits with intact skin were used as test animals. The animals were acclimatised and cared for as specified in ISO 10993-2, approved 25 June 2019 [37] and Ordinance Nr. 20 (State Gazette of Bulgaria, Nr. 87, 09.11.2012). The fur on the back of the animals was carefully clipped (10 × 15 cm) 4 h before the test. The general scheme of the skin application sites is presented in Appendix A. Two microneedle strips (RV-alginate MN and RV-alginate/k-CRG MN) were placed directly on skin positions 1 and 4, respectively. The positive control (500 µL of 10% SDS solution) was applied on site 2, whereas a referent product (Exelon^®^) was applied as a control on site 3. All four fields were covered with sterile absorbent gauze (2.5 × 2.5 cm) and Omnifix^®^E (hypoallergenic adhesive non-woven tape) for 4 or 24 h (Appendix A).

The reaction of each application site was recorded at the 1st, 24th (±1), 48th (±1), and 72nd (±1) hour after removing the test sample, the covering gauze, and the Omnifix^®^E. Skin reaction was described and scored live for erythema (reddening of the skin or mucous membrane) and/or oedema (swelling due to abnormal infiltration of fluid into the tissues) according to the scoring system given in ISO 10993-10 (Appendix A). In parallel, the skin reaction was photo-documented (DSC-WX9 digital camera, 16.2 MP, Carl Zeiss^®^ Vario-Tessar^®^ Objektiv, Sony, Minato, Tokyo, Japan). The Primary Irritation Score (PIS) for each sample was calculated by dividing the sum of all scores by the number of animals (two test/observation sites, three-time points). The Primary Irritation Index (PII) was calculated based on the PIS, whereby all primary irritation scores of the individual animals were added and divided by the number of animals (generally three) (Appendix A). The cumulative Irritation Index (CII) is equal to the total number of skin reactions divided by the number of animals. Only the observations on the 24th, 48th, and 72nd hours were used for calculations.

### 2.7. Statistical Analysis

Statistical analysis was performed using GraphPad Prism Software (Version 7 for Windows; GraphPad Software Inc, San Diego, CA, USA). The two-way ANOVA analysis of variance was used to assess the differences between formulations at all timepoints for the % permeation. One-way ANOVA analysis of variance was used to assess the differences between formulations at each timepoint of Apparent Permeability, % retained in the skin, and % remaining apical. One-way ANOVA analysis was also considered for the texture analysis results.

## 3. Results and Discussion

### 3.1. Optimisation of Microneedle’s Production

Polymeric MNs were prepared using silicone templates, and two polymers were considered to produce three different hydrogels based on their characteristics, namely, viscosity and mechanical properties in the dry state. Comparing the different formulations produced, alginate 7% (*w*/*w*) appears to be a more viscous hydrogel than k-CRG 1% (*w*/*w*), while the mixture of the two hydrogels showed an intermediate profile (Appendix A). MN production was optimised with calcein-loaded hydrogels to define an efficient and reproducible production protocol. Based on the literature, Method 1 was implemented (Table 1) [32]. However, using this approach, significant polymer loss was noticed, suggesting the amount of polymer added to each template was exceeding that supported by the moulds used in the present study. In addition, the centrifugation cycles were time-consuming, and therefore all these factors were adjusted, resulting in Method 2 (Table 1). The temperature of centrifugation was also decreased, expecting to have a less fluid hydrogel and avoid the escape of the formulation from the moulds. However, in this approach, it was observed that a polymer was lost in the very first layer. Then, a different methodology was designed (Method 3, Table 1), in which the quantity of polymer added in the beginning was reduced, and additional incubation and centrifugation steps were included. However, this method generated MNs with no visible needles, which suggests the importance of the high centrifugation cycles, mainly in the first layer.

Method 4 considered longer centrifugation steps and a higher number of hydrogel layers. Moreover, a higher temperature (30 °C) was set during the centrifugation process, similar to Method 1. Even though the temperature may influence the fluidity of hydrogels, in the literature, there is no information regarding the eventual relevance of this parameter on the production of MNs. Most often, the temperature used when producing the devices is not mentioned [38,39,40]. This method allowed the successful production of the devices since no evidence of polymer loss was detected, and visible needles were observed even with naked eyes. Thus, Method 4 was considered for the optimised protocol for MN production.

#### 3.1.1. Morphological Analysis

The morphology of each MN type made from different formulations was evaluated by SEM analysis for drug-free MNs (Appendix A). Both alginate and alginate/k-CRG mixture MNs presented a three-dimensional (3D) pyramidal structure with sharp tips. However, comparing both formulations in the different sizes, 800 µm arrays (Appendix A) presented MNs with a more pronounced curvature, especially in the first row. These slight defects can be caused either by the height of the MNs since bigger might be more flexible/fragile or by the detachment process itself. In fact, after the drying process, the arrays were separated from the mould using duct tape to grab onto the MN system and pull the MNs from one corner to the other, possibly causing the curvature observed in the first rows of MNs. The size of the obtained needles agrees with the manufacturer’s information since 400 µm moulds have generated needles of approximately 373 µm height. Regarding the 800 µm moulds, they allowed the formation of 619–643 µm height. The difference from the expected and obtained size may be due to the drying process of the MNs and the loss of volume by water evaporation, a fact that seems to be more pronounced for 800 µm devices, as previously reported [41]. Nevertheless, differences between the expected and obtained sizes correspond to 7 and 23% of the size, respectively, for 400 and 800 µm devices; values do not compromise the intended transdermal delivery.

Overall, alginate and alginate/k-CRG hydrogels allow the preparation of the MNs following the optimised methodology resulting in devices with the desired morphology (Appendix A). As opposed, k-CRG hydrogel (Appendix A) did not allow the formation of MN presenting pyramidal structure for both MN heights. Therefore, k-CRG MNs were discarded from further studies given the absence of the characteristics to puncture the skin.

Using calcein as a model compound, 400 and 800 µm MN arrays were produced following the optimised conditions using alginate and alginate/k-CRG mixture hydrogels. The obtained devices presented the desired 3D pyramidal structure, and the size of the needles agreed with the expected values, varying from 356 to 361 µm and 620 to 720 µm, respectively, for devices produced using 400 and 800 µm moulds. As verified for drug-free MNs, some needles of the first row exhibited a curvature, as shown in Figure 2A. As found for non-loaded MNs, a slight curvature of the needles can be observed for calcein-loaded MNs, mainly for 800 µm alginate/k-CRG MNs (Figure 2D). Taking into account that the curvature was verified for drug-free and calcein-loaded MNs, the cause may be associated with manipulation of the needles during detachment or by the weight of the structure. Although some defects in the 3D structure of these devices were perceptible, in general, they appear to have suitable morphology for skin permeation studies.

#### 3.1.2. Texture Analysis

For an MN to penetrate the skin effectively and deliver the therapeutic drug, the force that is applied should not deform or break the needles; thus, in this study, the compression force was investigated. The profiles of force (F) versus displacement after an axial force load are represented in Appendix A. A general idea of which type of MN offers the best resistance to force can be drawn by analysing these profiles. Table 2 summarises the maximum force detected by the probe at 0.3 mm of distance. Results allowed some conclusions regarding the influence of the presence of the model drug as well as the impact of the hydrogel composition and the height of the MN on the compression force. The reading of a reduced set of needles by the probe, as opposed to the overall array at once, acknowledges the heterogeneity factor of the device since it may be possible that not all needles have the same strength [40].

Alginate MNs exhibited compressive forces superior to those experienced by alginate/k-CRG MNs under all studied conditions. Regarding the height, results have shown that drug-free MNs and calcein-loaded MNs have a compression force superior in 800 µm MNs compared to 400 µm MNs. The presence of calcein affected the compression force of alginate MNs, contrary to what happens with alginate/k-CRG MNs, which does not present a significant difference compared to the respective drug-free MNs. For the developed MNs, the force/needle values ranged from 0.23 to 1.01 N, which were indicative of the MNs’ ability to enter the skin. In fact, for the production of a robust MN array capable of piercing the skin, the failure force *per* needle needs to exceed the force required to successfully insert the MN into the skin. Yu and colleagues [42] produced a patch containing MNs produced with alginate and hyaluronate able to surpass the *SC* since the fracture force experienced by the needles produced in the study was inferior to 0.098 N/needle [42]. Additionally, Donnelley et al. have suggested that an insertion force as low as 0.03 N/needle may be enough to pierce the *SC* [43].

#### 3.1.3. Permeation Assay

A porcine skin permeation assay was performed to characterise the transdermal permeation profile of the four MN types (Figure 3 and Appendix A). Skin permeation of calcein increases over time, reaching a maximum percentage of approximately 40% after 24 h for all studied conditions. For short-term incubation, 3 h, no significant permeation was observed. Differences between the two delivery systems, hydrogel versus MNs, were observed after 6 and 8 h of the assay. Namely, alginate MNs have superior calcein skin permeation than their hydrogel counterpart, while alginate/k-CRG MNs had lower calcein skin permeation compared to the equivalent hydrogel (Appendix A). The fact that there were differences between hydrogel versus MN demonstrates that these systems, when implemented, induce changes in the skin that influence compound absorption. Alginate MNs have shown a faster release over time, whereas the mixture presented a smaller percentage of permeation when compared to the respective hydrogel (Appendix A).

Considering the four types of MNs after 6 and 8 h skin permeation (Figure 3A), alginate MNs and 800 μm alginate/k-CRG MNs exhibited the highest percentage of permeation than alginate/k-CRG MNs while 400 μm alginate/k-CRG MNs at 8 h presented a significantly lower permeation rate.

Nevertheless, in general, both sizes presented a similar release profile of the model drug over time (*p* > 0.05). Data are in agreement with the values obtained in the apparent permeability (*P_app_*) calculations (Appendix A). Shakel and co-workers also reported the study of calcein permeation using pig ear skin with the *P_app_* values after 3 h of the assay in the same order of magnitude of 10^−^^6^ cm/s (Appendix A) [35].

Even though approximately 40% of calcein was able to cross the skin barrier, the remaining 60% was either retained in the skin or remained in the apical compartment of the Franz Diffusion Cells. Figure 3B and Appendix A show the overall scenario of calcein distribution in the permeation assay system. Despite the lack of statistical significance between the calcein-loaded MNs of different needle sizes, and having in mind a systemic application for RV, 800 µm MNs were selected to proceed with this study as they are expected to overcome the epidermis (ca. 150 mm) and reach the dermis with a sustained permeation profile [44].

#### 3.1.4. MNs Interaction with the Skin Assessed by Microscopy

Upon skin permeation assay with the developed transdermal delivery systems, the skin was processed for visualisation under optical and confocal fluorescence microscopy. For the sample in contact with free calcein, a “loose” dermal region was visible in Figure 4A in comparison to skin treated with PBS, considered as a control (Appendix A). The transition site between skin with long calcein contact and the untreated skin is marked with a dashed circle (Figure 4A). This apparent degradation was also observed in samples containing both types of hydrogel formulations (Figure 4B,C), showing similar morphological aspects. This indicates that the penetration made by calcein under these conditions may affect the skin structure and function.

In contrast to all other conditions, Figure 4D concerns a skin sample not subjected to a permeation assay. Having been in contact with an MN array for 10 min, the darker pink areas may indicate the presence of a needle (marked with arrows). Given the short period of time that the system was inserted into the skin, possible needle marks are more difficult to observe. Consequently, the remaining conditions represented an interval of exposure of 24 h.

Figure 4 portrays skin samples subjected to different MN arrays. All skin samples present a darker structure resembling the presence of an MN that punctured the skin (highlighted in higher magnification, Figure 4(d1),e–h. The lengths of the structures appear to be superior to the actual size of an MN height expected considering the morphological characterisation.

The site where an MN has been inserted can be identified by the triangular shape left on the skin surface due to the MN base, as well as the change in colouration of the area (darker areas). However, the measurement of this structure may not be faithful to the pore created by the MN as the skin may experience some folds or breaks during the processing of the skin for analysis. Therefore, the measurement of the puncture structures does not allow conclusions to be taken as to the height of the MN used. Nevertheless, the dimensions of the MNs evaluated in the SEM analysis corroborate the ability of the developed systems to deliver the drug since passage through the *SC* only requires an MN length of 10–20 µm in humans [45]. Moreover, both MN sizes allow the delivery, in a first instance, to the dermis (1–2 mm) [46], which is in accordance with the goal of systemic delivery of the therapeutic drug.

Of note, some of the skin sections display folds that could be mistaken for skin breakdown caused by MNs (Figure 4B,D,E,H marked with *). However, this may be due to some artefacts caused by the skin manipulation in the preparation of the histological samples, given the small thickness of the skin sections caused by the handling of the sample when mounting it on the slide.

Skin samples were also analysed after permeation assays using confocal fluorescent microscopy (Figure 5). The skin incubated with calcein solution also exhibited a “loose” appearance (as depicted in Figure 4A), suggesting that decomposition of skin components has occurred, which would therefore increase skin permeability (Figure 5A). This event was proven to be caused by contact with calcein as the surrounding skin (no contact with calcein solution) appears unchanged. Since there was the degradation of the skin in the environment containing calcein solution, the calcein signal is expected to be less intense once most of the compound permeated to the basolateral compartment. Regarding calcein-loaded hydrogel formulations, alginate, and alginate/k-CRG hydrogels, skin samples also present the “loose” appearance of the skin reported for the calcein solution sample (Figure 5B,C).

For MN-treated skin samples, the array remained in direct contact with the skin for 10 min. Figure 5D shows the skin sample processed after MN array removal and was possible to observe, marked by red arrows, disruption of the *SC* caused by the perforation of an MN. All MN skin samples appear to have invaginations on their surface (Figure 5E–H, evidenced by red arrows). The green colour represents the unspecific fluorescence caused by the calcein and can be related to the intensity of the calcein signal. The control sample (skin treated with PBS without calcein) was visualised, and no fluorescence was detected (data not shown). The signal was morphologically less intense for 800 µm MNs (Figure 5F,H) compared to 400 µm MNs (Figure 5E,G), suggesting a potential for systemic drug delivery since the latter devices may allow higher skin retention of the model drug.

### 3.2. Application of the Optimised Microneedles for Rivastigmine Transdermal Delivery

#### 3.2.1. Morphological and Mechanical Characterisation

RV-loaded MNs (800 µm) were produced with alginate and alginate/k-CRG compositions, and the morphological evaluation by SEM is displayed in Figure 6A,B. Both types of MNs retain a 3D triangular pyramid shape in the presence of the anti-dementia drug. However, the RV-loaded alginate MN device showed slightly thinner and elongated structures with curved tips, which can compromise the mechanical strength of the array and, subsequently, the skin perforation (Figure 6A). This effect was not observed under the same conditions for calcein-loaded MNs (Figure 2B), suggesting the alteration of MNs morphology with the addition of this RV, even though both studied molecules present similar hydrophilicity and low molecular weight [5,47]. Alginate/k-CRG MNs present a more uniform structure without any defects (Figure 6B). Moreover, when comparing both RV-loaded MNs with their respective drug-free MN (Appendix A), the overall structure seems unchanged. MNs height ranged from 680 to 770 µm, as previously observed for drug-free and calcein-loaded MNs (Appendix A and Figure 2, respectively).

The mechanical properties of RV-loaded MNs were investigated through the profile of force versus displacement after the force load inflicted onto the MNs (Appendix A) and the mechanical strength (Figure 6C). From the compressive force data, it was possible to infer that the presence of RV influenced the mechanical properties, as already reported for other compounds [48,49]. RV-loaded alginate MNs experience an increase in compressive strength by more than two times when compared to drug-free alginate MNs, reaching the 5.74 N of force per needle. In sum, RV-loaded alginate MNs have approximately 10 times the compression force of alginate/k-CRG MNs, but both experience a compression force per needle able to cross the *SC* [42].

The incorporation of compounds into an MN system may alter the compression force experienced. Bhatnagar and collaborators [48] developed a dissolving polymeric MN array based on polyvinyl alcohol and polyvinyl pyrrolidone for the delivery of an antibiotic, besifloxacin, through the cornea. The texture characterisation, using axial and transverse force, showed a decrease in compression force of besifloxacin-loaded MNs of 7.64 N when compared to 8.80 N for drug-free MNs [48]. Another study, based on the transdermal delivery of a compound normally applied intradermally, tetanus toxoid (TT), demonstrated the reduction of the maximum force for a displacement of 0.5 mm for TT-loaded MNs compared to blank MNs [49].

#### 3.2.2. Skin Permeation Assay

Transdermal delivery of RV was evaluated using a porcine skin permeation assay, having an Exelon^®^ patch and both hydrogel formulations as controls. The Exelon^®^ patch 5 cm^2^, available in the market for transdermal delivery of RV, allowed us to compare the optimised MNs with a commercially available form, but whose drug bioavailability and irritability are not the most favourable. Given an effective drug permeation observed for all studied conditions, reaching maximum values of approximately 100% after 24 h, Figure 7A focuses on the first 8 h of study.

All the prepared RV-loaded formulations and MN devices lead to more than 50% RV permeation, yet lower values than that observed with the Exelon^®^ patch (Figure 7A). Particularly after 8 h of the assay, there were significant differences in the transdermal delivery of RV through the skin (*p* < 0.05) between the MN system and the RV-loaded hydrogel made from alginate (0.43 and 0.45 mg of permeation, respectively) (Figure 7A and Appendix A) vs. the commercial patch (0.80 mg). Although not significantly different (*p* > 0.05), the same can be observed for alginate/k-CRG-based hydrogels and MNs (0.66 and 0.49 mg of permeation, respectively) (Figure 7 and Appendix A). For all the studied formulations containing RV, the release kinetics followed a zero-order pattern (r^2^: 0.951, 0.977, and 0.987, for physiological, RV-alginate MN, RV-alginate/k-CRG MN, and Exelon^®^, respectively). The zero-order model describes a release rate independent from drug concentration. Moreover, Appendix A shows that r^2^ values were found for each formulation and commonly studied kinetic models. In sum, the RV level at the site of action remains constant throughout drug delivery once administered either as incorporated within a polymeric hydrogel or commercial patch, independent of its concentration.

After skin permeation studies, both apical content and pig ear skin were collected, and drug quantification was performed ( Figure 7B and Appendix A). Since the RV loaded in the donor compartment of Franz cells permeated completely, the amount of drug remaining in the other components of the system would be residual (a maximum of 0.045 mg of RV retained after 24 h), according to the high permeation values discussed above. The results depicted in Figure 7B supported the conclusions reached. The majority of RV permeated the skin, while a small percentage remained in the apical or was retained in the pig skin. These results confirm the potential of both RV-loaded MNs to be a sustainable TDD system for a systemic release of the drug. The developed MNs could be feasibly administrated for a period longer than 24 h.

In clinics, the therapeutic dose of RV corresponds to 50% of the amount of drug-loaded in the patches [10]. Under the described skin permeation assay condition, the Exelon^®^ patch reached this goal after 6 h, while the developed hydrogels and MN devices required 8 h, suggesting a more controlled release profile for the polymeric systems. The permeation values were in agreement with the data obtained for the Papp calculations, as shown in Appendix A. No statistical significance (*p* < 0.05) was observed throughout these results. However, the same tendency has been detected in which the produced hydrogels and MNs presented a slow rate of diffusion over time than that verified for the Exelon^®^ patch.

A study elaborated by Chauhan and Sharma [11] compared the delivery of RV between Exelon^®^ patch and optimised formulations of RV-loaded NLC integrating either Eudragit E-100 or poly-butyl methacrylate-co-methyl methacrylate onto the patch. An in vitro drug release study performed demonstrated that drug release was faster for the Exelon^®^ patch, with a percentage of 95.68% of release after 24 h, while the nanocarriers patch matrix only reached the same percentage after 72 h, beneficial for drug delivery systems looking for the controlled release of the therapeutic formulation [11]. Another approach resorted to the delivery of RV using poly (lactic-co-glycolic acid) NPs for intranasal delivery embedded in a poloxamer 407^®^ matrix [50]. Drug release studies using the dialysis bag diffusion technique compared three different formulations of NPs with the respective RV-loaded hydrogel. Drug release was comprised for NPs between 43.28% and 60.41%, while hydrogel formulation reached 62%. Although the gel formulation presented higher release, the profile of release of gel formulations containing NPs showed a more controlled release explained by the barrier RV must go through to be delivered. The results are in concordance with the results shown above. However, it is important to mention that in both referred works, the authors only evaluated in vitro drug release studies instead of considering skin permeation investigations.

#### 3.2.3. Skin Irritation Test

Microneedles are considered minimally invasive devices, but their safety application on the skin should be examined. For example, Xing et al. performed a skin irritation test of microneedles based on another type of polysaccharide microneedle and found that sodium carboxymethyl cellulose did not provoke irritation reactions on rabbits’ skin [51]. Of note, the use of a transdermal patch, such as Exelon^®^, requires contact of a patch with the skin for 24 h, which may trigger a negative response by the skin. Aspects such as the nature of the drug, the contact area, the existence of a history of allergy to any of the components, skin lesions, or duration of contact appear as risk factors for the development of skin reactions. A study conducted to evaluate skin tolerability to Exelon^®^ patches of 10 cm^2^, and 20 cm^2^ reported that most of the skin reactions were considered “slight” or “mild” and consisted of pruritus and erythema, while more severe responses only concerned a small percentage of the patients under study (3–7%) [12].

Considering these facts, in the present study, dermal effects of the developed MNs were evaluated and compared with that of Exelon^®^ (commercial product). RV-alginate MNs and RV-alginate/k-CRG MNs arrays were applied for 4 h and 24 h on rabbits, and their dermal reaction was assessed at the 1st, 24th, 48th, and 72nd hours after the exposition period (Figure 8A and Appendix A). After 4 h of contact, the PIS and PII for RV-alginate MN (position 1) and RV-alginate/k-CRG MN (position 4) were equal to zero (Appendix A). PIS and PII were also zero for the referent product. The small wound that was noticeable at position three was caused by the removal of the strip. In comparison, erythema was registered after the treatment with the positive control (10% SDS) at the 24th, 48th, and 72nd hours. As far as a single exposition of alginate and alginate/k-CRG MNs did not lead to skin irritation, a CII was not calculated.

Both types of MN strips were applied for 24 h on rabbits and their dermal reaction was assessed at the 1st, 24th, 48th, and 72nd hours after the exposition period (Figure 8A). PIS and PII for RV-alginate MN (position 1) and RV-alginate/k-CRG MN (position 4) was equal to zero (Figure 8B). PIS and PII were also zero for the referent product (Figure 8B). The small wound that was noticeable at position three was due to the strong adhesion and its difficult detachment after 24 h; it was slight mechanical damage rather than a skin reaction. As seen in the next observations (24, 48, and 72 h), there was no spreading beyond the area of application, which proves that it was not an irritation effect. Only the treatment with the positive control (10% SDS) resulted in skin irritation reactions. As seen, the PIS and PII of the positive control (10% SDS) increased to values four and two, regarding the erythema and oedema reactions, respectively (Figure 8B). As far as the single exposure of RV-alginate and RV-alginate/k-CRG, MNs did not lead to skin irritation; a CII was not calculated. Thus, the overall results have shown that both test samples, RV-alginate and RV-alginate/k-CRG MNs, did not cause skin irritation after 4 or 24 h of exposition. Moreover, the findings proved that the developed MNs were innocuous upon removal after 4 or 24 h, in contrast with the evidence found for Exelon^®^, the commercial patch, which caused slight mechanical damage to the skin.

## 4. Conclusions

This work demonstrated that MN arrays obtained using alginate or alginate/k-CRG mixture enable efficient delivery of RV through the skin. A micro moulding-based production method was optimised to obtain polymeric MNs using calcein to allow a more detailed characterisation of the arrays. MNs counteract the compressive deformation of the skin and provide mechanical strength during the application, allowing complete insertion of the devices. Polymeric RV-loaded MNs (800 µm) were evaluated for TDD, exhibiting that approximately 0.45 mg of RV permeated the skin over 8 h. Skin irritation assay supported their safe use about commercially available Exelon^®^. Overall, results pointed out that both polymeric formulations can be considered to produce RV-loaded MNs, thus presenting a simple, cheap, efficient, and safe approach to delivering RV. To the best of our knowledge, no other k-CRG-derived MNs have been produced, nor another polymeric device for RV delivery, and only a hollow MN has been reported. In conclusion, this study indicates that polymeric MNs loaded with RV may represent a future alternative to its administration, contributing to improving patients’ quality of life.

## Figures and Tables

**Figure 1 pharmaceutics-14-00752-f001:**
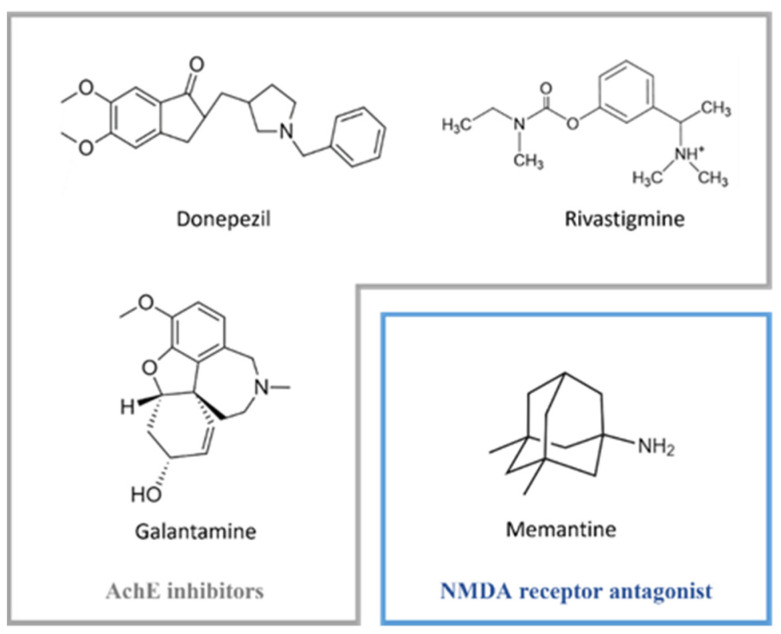
Examples of AchE inhibitors (grey box) and NMDA receptor antagonist (blue box) currently used in pharmacotherapy for neurodegenerative dementia.

**Figure 2 pharmaceutics-14-00752-f002:**
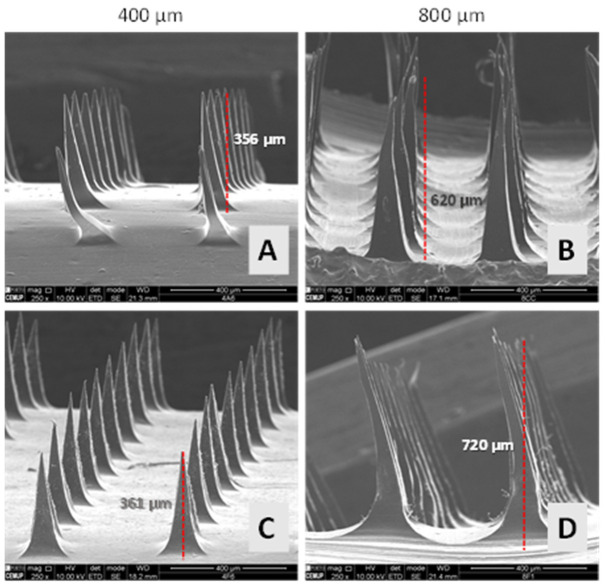
Representative SEM images of 400 and 800 µm calcein-loaded MNs. (**A**,**B**) alginate MNs; (**C**,**D**) alginate/k-CRG MNs.

**Figure 3 pharmaceutics-14-00752-f003:**
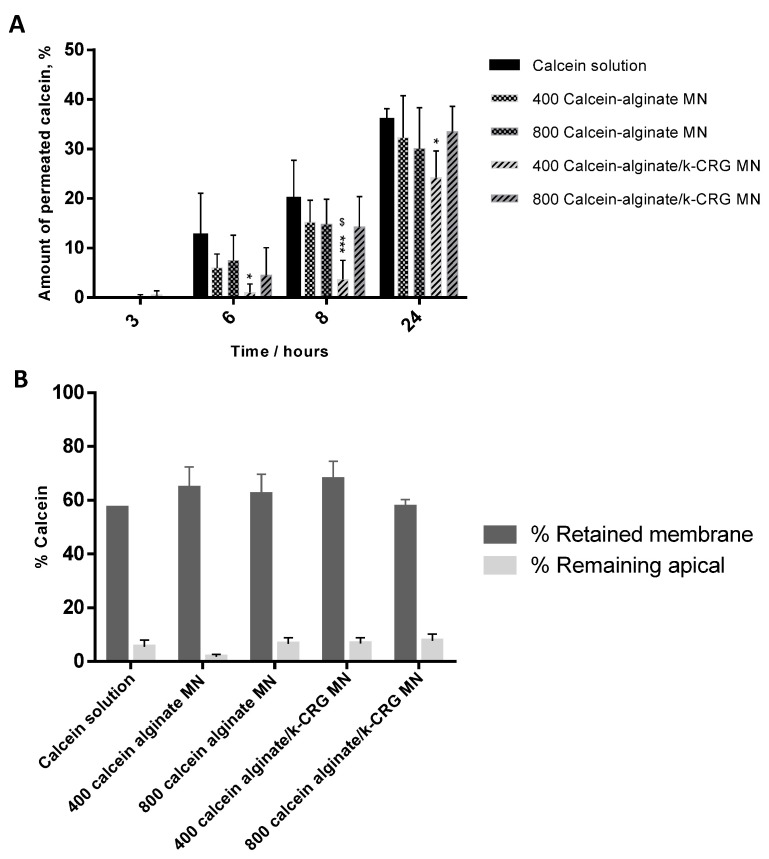
Calcein permeation profile. The bars/points represent the mean ± SD of at least three independent experiments (*n* = 3). (**A**) Amount of permeated calcein for porcine skin treated with calcein solution and calcein-loaded MNs of two different heights, at different time points. * *p* < 0.05 for 400 calcein-alginate/k-CRG MN vs. calcein solution at 6 h; *** *p* < 0.001 for 400 calcein-alginate/k-CRG MN vs. calcein solution at 8 h; ^$^ *p* < 0.05 for 800 calcein-loaded alginate/k-CRG MN vs. 400 calcein-alginate/k-CRG at 8 h; * *p* < 0.05 for 400 calcein-alginate/k-CRG MN vs. calcein solution at 24 h. (**B**) Distribution of calcein among skin retained and non-permeated through the porcine skin after 24 h.

**Figure 4 pharmaceutics-14-00752-f004:**
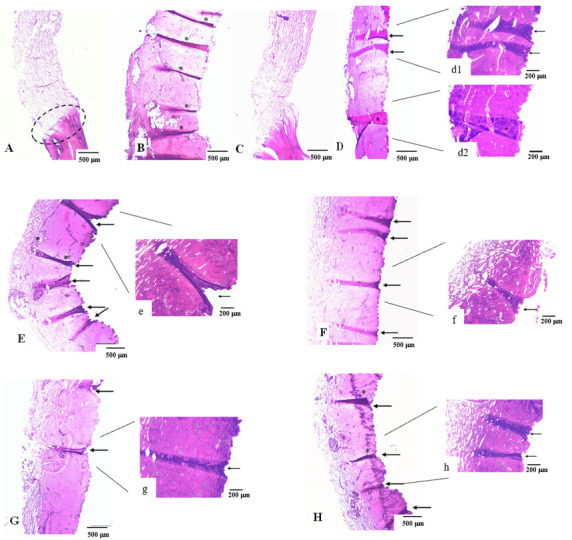
Optical microscopy analysis of skin samples from the permeation assay. The skin was stained with H&E upon 24 h exposure to (**A**) calcein solution; (**B**) calcein-alginate hydrogel; (**C**) calcein-alginate/k-CRG hydrogel; (**E**,**e**) 400 calcein-alginate MN; (**F**,**f**) 800 calcein-alginate MN; (**G**,**g**) 400 calcein-alginate/k-CRG MN; (**H**,**h**) 800 calcein-alginate/k-CRG MN. Samples (**D** (**d1**,**d2**)) corresponds to pig ear skin after 10 min in contact with 800 calcein-alginate/k-CRG MN. Amplified regions of the conditions are marked with the equivalent small letter. In all images, the epidermis is displayed oriented to the right. Folds due to skin manipulation in the preparation of the histological samples are indicated by the symbol *. Scale bar 500 µm for Figures (**A**–**H**) and 200 µm for amplified regions (**d1**,**d2**,**e**–**h**).

**Figure 5 pharmaceutics-14-00752-f005:**
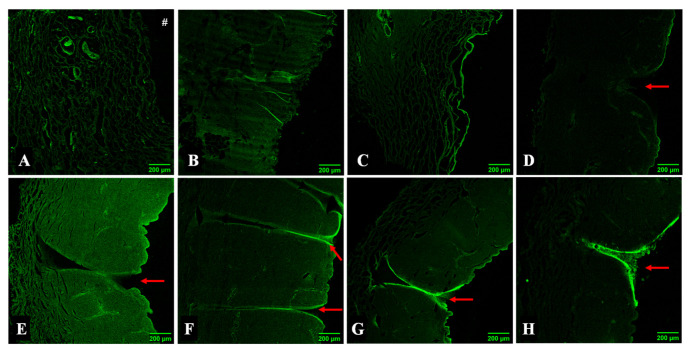
Confocal fluorescence microscopy analysis of skin samples resultant from the permeation assay. Skin treated for 24 h with (**A**) calcein solution; (**B**) calcein-alginate hydrogel; (**C**) calcein-alginate/k-CRG hydrogel; (**D**) 800 calcein-alginate/k-CRG MN for 10 min; (**E**) 400 calcein-alginate MN; (**F**) 800 calcein-alginate MN; (**G**) 400 calcein-alginate/k-CRG MN; (**H**) 800 calceinalginate/k-CRG MN. The # symbol represents the orientation of the epidermis. The red arrow indicates the structures possibly created by the MNs. Scale bar 200 µm.

**Figure 6 pharmaceutics-14-00752-f006:**
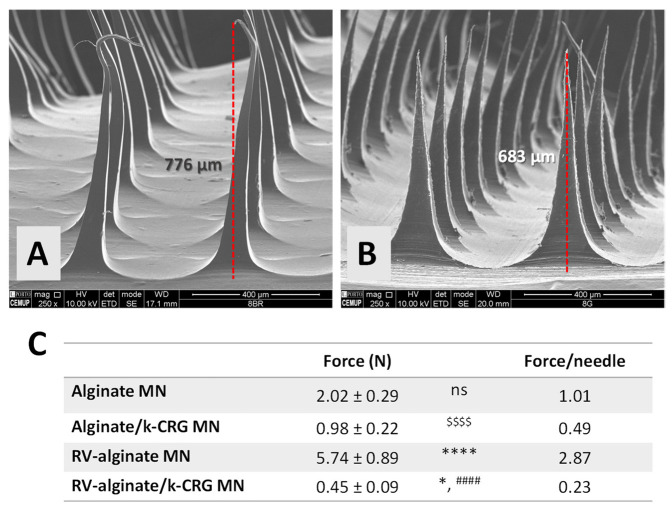
Structural and mechanical characterisation of RV-loaded MNs. Representative SEM images of RV-loaded (**A**) alginate and (**B**) alginate/k-CRG MNs. (**C**) Profile of the mechanical forces of drug-free and RV-loaded MNs. Data represented are the mean and SD of all values measured. * *p* < 0.05 for RV-alginate/k-CRG MN vs. alginate/k-CRG MN; **** *p* < 0.0001 for RV-alginate MN vs. alginate MN; ^####^ *p* < 0.0001 for RV-alginate/k-CRG MN vs. RV-alginate MN; ^$$$$^ *p* < 0.0001 for alginate/k-CRG MN vs. alginate MN. ns—no statistical significance (*p* > 0.05).

**Figure 7 pharmaceutics-14-00752-f007:**
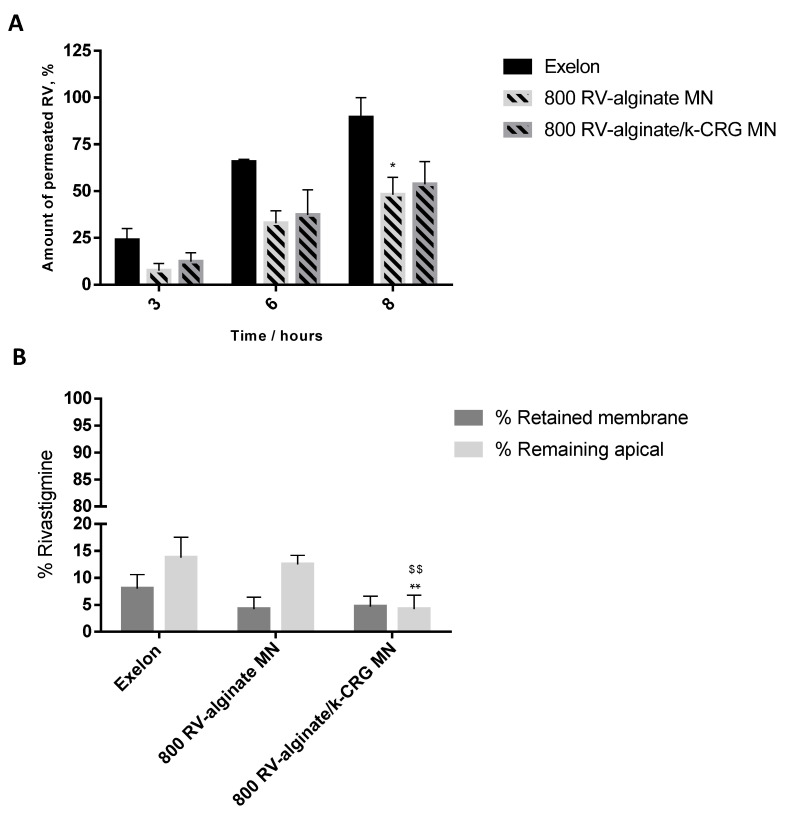
Skin rivastigmine permeation profile. The bars/points represent the mean ± SD of the permeability for three independent experiments (*n* = 3). (**A**) Amount of permeated RV (%) as a function of time obtained for Exelon^®^ and RV-loaded MNs. * *p* < 0.05 for 800 alginate MN vs. Exelon^®^ at 8 h. (**B**) Distribution of RV retained in the porcine skin and remaining in the apical compartment. ** *p* < 0.01 for 800 RV-alginate/k-CRG MN vs. Exelon^®^ remaining in the apical; ^$$^ *p* < 0.01 for 800 RV-alginate/k-CRG MN vs. 800 RV-alginate MN remaining in the apical.

**Figure 8 pharmaceutics-14-00752-f008:**
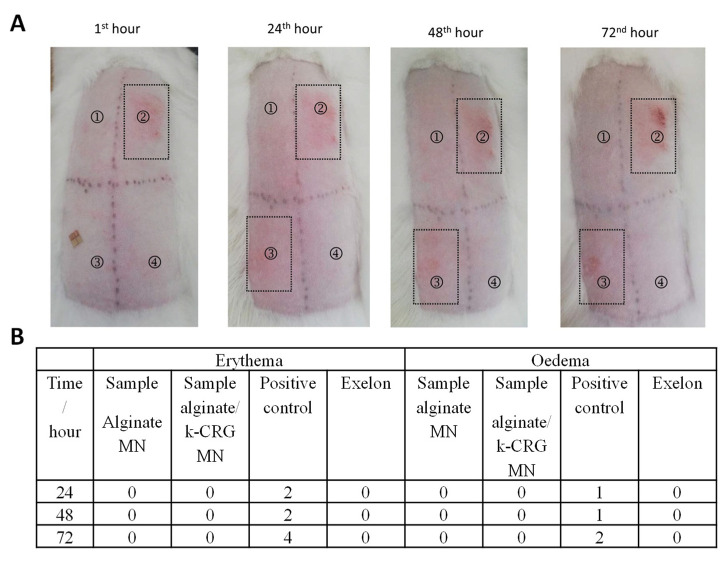
Skin irritation test for RV-alginate and RV-alginate/k-CRG MNs after 24 h exposition. (**A**) digital photography: 1—RV-alginate MN; 2—positive control (10% SDS); 3—Exelon^®^ patch; 4—RV-alginate/k-CRG MN. (**B**) Evaluation of 24 h exposition following the scoring system for skin reaction.

**Table 1 pharmaceutics-14-00752-t001:** Summary of MN production method optimisation.

		Method 1	Method 2	Method 3	Method 4
Number of layers	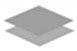 2 layers	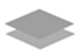 2 layers	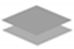 2 layers	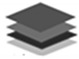 4 layers
Time	3 days	3 days	3 days	3 days
Conditions per day	Day 1	50 mg; 3500 rpm; 30 °C; 1 h	100 mg; 2500 rpm; 30 °C; 30 min	50 mg; 500 rpm; 15 °C; 15 min	25+25 mg; 3500 rpm; 30 °C; 1 h
	Day 2	150 mg; 3500 rpm; 30 °C; 1 h	50 mg; 500 rpm; 15 °C; 15 min	100 mg; 500 rpm; 15 °C; 15 min	50 mg; 3500 rpm; 30 °C; 1 h
	Day 3	Incubation; 100 rpm; 25 °C; 3 h	Incubation; 150 rpm; 25 °C; 3 h	Incubation; 150 to 250 rpm; 25 °C; 2 h	50 mg; 3500 rpm; 30 °C; 1 h
Centrifugation; 500 rpm; 10 °C; 1 h
Incubation; 150 rpm; 25 °C; 3 h	Incubation; 150 rpm; 25 °C; 3 h
Observations	Visible Polymer loss. Possible overfilling of the micromould	Polymer loss detected after first-layer centrifugation	After detachment, no picks were visible	No significant polymer losses. Needles visible to the naked eye

**Table 2 pharmaceutics-14-00752-t002:** Profile of the compression forces of non-loaded and calcein-loaded MNs.

Type of MN	Force (N) ^1^	Force/Needle
400 µm alginate MN	1.02 ± 0.20	0.51
800 µm alginate MN	2.02 ± 0.29 ****	1.01
400 µm Calcein-alginate MN	1.30 ± 0.25 ^#^	0.65
800 µm Calcein-alginate MN	1.65 ± 0.32 ***^, ###^	0.83
400 µm alginate/k-CRG MN	0.52 ± 0.12	0.26
800 µm alginate/k-CRG MN	0.98 ± 0.22 ****	0.49
400 µm Calcein-alginate/k-CRG MN	0.45 ± 0.07	0.23
800 µm Calcein-alginate/k-CRG MN	0.92 ± 0.16 ****	0.46

^1^ Data represented is a mean and SD of all values measured for force values. *** *p* < 0.001 for 800 calcein-alginate MN vs. 400 calcein- alginate MN; **** *p* < 0.0001 for 800 calcein-alginate/k-CRG MN vs. 400 calcein- alginate/k-CRG MN; **** *p* < 0.0001 for 800 alginate MN vs. 400 alginate MN; **** *p* < 0.0001 for 800 alginate/k-CRG MN vs. 400 alginate/k-CRG MN; ^#^ *p* < 0.05 for 400 calcein-alginate MN vs. 400 alginate MN; ^###^ *p* < 0.001 for 800 calcein-alginate MN vs. 800 alginate MN.

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
