# Peer review of "Polymeric Microneedles for Transdermal Delivery of Rivastigmine: Design and Application in Skin Mimetic Model"

_pharmaceutics, 2022, doi:10.3390/pharmaceutics14040752_

Round 1

Reviewer 1 Report

Dear Authors,

This paper is an interesting work, well writen, the results being supported by an extensive experimental part, with valuable conclusions concerning the perspective of development of new delivery systems for transdermal release of rivastigmine, using the microneedles technology in the dementia patients treatment, improving their life quality.

I have some minor remarks related to:

  1. As a suggestion, the Authors could model the permeation data, setting up a kinetic model.
  2. Some minor English revision and spelling corrections are required.

Reviewer 2 Report

Dear Authors

The work presented here is of high quality. Bibliography gives a very interesting approach to the drug delivery options and possibilities and the experiments with ND in comparison with patches are very well defined. Instrumental determinations are enough and accurate for to deliver the information needed

I consider, after several readings that the paper is ready to be published as it is ºnow.

Many thanks

Reviewer 3 Report

The article by Guimaraes, T.M.T. et al. about the transdermal (or transepidermal) delivery of rivastigmine with microneedles appears scientifically correct and well organised. Authors performed a nice experiment, but sadly the histological study is terrible.

The text has a generally decent grammar / style, lacking commas in various sentences. There are some grammar issues, some of them mentioned as minor faults. The style may also be improved. For this reason, a grammar / style revision (not limited to the mentioned issues) would improve the manuscript, and is recommended.

There are major issues:

  • Abstract could be more dynamic and coherent. I think some ways are: mention briefly the limitations of both mentioned routes; include less information about the methods and the experiment (highlighting the most relevant data and removing some less important aspects); I also feel confusing the jump from rivastigmine to calcein without a proper explanation of the role of calcein.
  • A notable fault in the whole work is that there are no references to the number of experiments performed.
  • I feel the histological pictures are really bad (Figure 4). The tissue seems to have folds due to a defective laboratory manipulation. All the photos are taken only with low magnification and also have low resolution (even the scale bars are difficult to read). You refer you performed Hematoxilin-Eosin, but I only identify eosin staining, can´t see anything stained with hematoxilin. Why do you include 2 scale bars? Thickness measures are confusing, as long as dermal thinckess can be altered if you retire previously the hypodermis; in fact, it is very possible to alter it when you retire the subcutaneous fatty tissue. You should contact with a properly trained histologist or pathologist (in any case with experience in cutaneous pathology), or even avoid the inclusion of this histological chapter.
  • Finally, did you perform any microscopic assessment with rivastigmine??? I only see calcein in this section. You can find various references about the microscopic effects of MN in the skin in the literature. If you want to be innovative, you should compare the histological effects of both employed compounds.

There are some minor issues as well:

  • Abstract, page 1, line 13. “MinimalLY”.
  • Abstract, page 1, lines 27-28. You should mention first the commercial name, and then you can refer it as “commercial patch”. All the future mentions to the name Exelon should have the registered symbol.
  • Introduction, page 2, line 55. This initial sentence mentioning Exelon should mention that they are rivastigmine patches.
  • Introduction, page 2, line 70. “Parenteral”.
  • Introduction, page 2, line 79. “, expanding”
  • Introduction, page 3, lines 109-110. Again, you should clarify the role of calcein. Was it employed to optimize the technique later applied to rivastigmine?
  • Materials, page 3, line120. What country is EUA? May be USA?
  • Materials, page 3, line 121. You should give more data about the pigs (number, age, gender, time before fixation…).
  • Materials, page 3, lines 127-129. You should be consistent with the language, it is strange for me to see one section of the “CEMUP” with the name in portuguese, and another section with the name in english.
  • Materials, page 3/4 , lines 334-345. Skin reaction was described and scored live or you made photographs to compare the different specimens later? In any case, I think you could describe the device you used to make the photos depicted in the figures.
  • Materials, page 4, section 2.2.2. Please clarify the role of calcein here too.
  • Materials, page 4, line 163. “Another three protocols”.
  • Materials, page 4, lines 165. The word “Following” seems to have no sense here. Please rewrite this sentence.
  • Materials, page 6, line 209. The “whole skin” also includes hypodermis. You should refer here to epidermis and dermal tissue.
  • Materials, page 6, line 213. You should give an idea about the exact measure.
  • Materials, page 8, line 308. CK2 is the model of an Olympus microscope.
  • Materials, page 8, line 310. Samples to analyse with confocal microscopy were HE stained or they were unstained? Did you use any antibody in the detection?
  • Materials, page 8, line 323. “Ethics”
  • Results, page 9, line 372. I find “However” suits better to the sentence than “Although”.
  • Results, page 10, line 391. “were separated”; please check the verb tense in all the manuscript.
  • Results, page 10, line 403. “As opposed, k-CRG…”.
  • Results, page 10, lines 411-413. Please, check this sentence.
  • Results, page 12, line 445. I think the sentence has a better sense if you change “superior for” for “superior in”.
  • Results, page 12, line 472. “significantly”. Lines 477-478. “The remaining % was”. Please consider a division of this paragraph. It´s too long.
  • Results, page 13, line 505. Figure 4C has no astersks. In the case you refer to figure 4B, these structures seems folds caused by a careless histological procedure, but without more detailed images, it is impossible to say.
  • Results, page 13, line 509-511. Darker pink areas are probably an artefact. If you magnify the image I could be more explicit about the cause of the artefact.
  • Results, page 14, lines 517-519. Under this magnification, the referred structures could be the London Bridge.
  • Results, page 14, line 521. “where a MN”
  • Results, page 14, line 540 (also 497). I don´t know what you mean with “lacy”. It is not a common histologic descriptor. May be you actually mean “loose”? If you include such a subjective extracellular matrix evaluation, you should include a normal (control) skin image for reference.
  • Results, page 15, lines 554-555. You should specify what antibodies are using (or whatever substance you are actually detecting). How do you know it is calcein?
  • Results, page 15, line 556. In the case you employed antibodies against calcein in the detection (which seems pretty reasonable), you can´t compare the intensity of fluorescence to determine the potential for drug delivery. The reaction is not precisely stoichiometric and image E has too background stain. I can´t see any significant differences in the other provided images (that have, by the way, very low resolution).
  • Results, page 15, line 557. “Their potential”.
  • Results, page 18, lines 640-641. This sentence needs proper citation.

Round 2

Reviewer 3 Report

The article by Guimaraes, T.M.T. et al. about the delivery of rivastigmine with microneedles was improved and is clearer now. The work still has some flaws, noted as major and minor issues.

The text has also an improved grammar / style, but authors should particularly check the commas in the revised version. Some grammar issues are mentioned as minor faults.

Major issues are:

  • It is still a fault in work the absence of a precise reference of the employed animals. I mean you should include a number of porcine ears, an age range and an estimation of sex distribution. You should not have difficulties to briefly include this information in materials & methods. You now indicate that the ears were fresh, but you should also give an estimation of how much time were in the fresh state before the formaldehyde fixation, for example was it a week, a pair of days, or less than 12 hours? You should also specify if the samples were freezed in any moment (I suppose they weren´t).
  • I feel the histological pictures of Figure 4 were notably improved. However, the low magnification is good to provide a panoramic view, but the details are still confusing. I encourage you to include some detailed images of the regions interpreted as MN remnants, and also of the regions interpreted as tissue folds. You can add a row of images below images E-H with higher magnification images of the slides.
  • Figure 5 would also benefit from the same image improvement performed in Figure 4, as long as the resolution is low, hindering the view of the scale bar.

In addition, I found some minor issues:

  • Abstract, page 1, line 22. “Were used”.
  • Introduction, page 2, line 82. “MNs, expanding the number of…”
  • Materials, page 8/9, lines 296-297 and 316-318. All samples were analysed.
  • Materials, page 9, line 326. I think the original unstained was better: “skin, unstained samples were…”
  • Results, page 14, line 519. I can´t find figure S7. May be there was a confusion with the supplementary file.
  • Results, page 14, lines 519-524. When you define this loose dermal region (actually it is not a “structure”), you should refer to this region with the same word. “The transition site between skin with long calcein contact and the untreated skin has been marked with a dashed circle (Figure 4A)”. “This apparent degradation was also observed in samples containing both types of hydrogel formulations (Figures 4B and 4C), showing similar morphological aspect”.
  • Results, page 14-15, lines 525-529, 531-534 and 539-541. I think you should describe the possible folds noted as asterisks together, to simplify the text. The darker dermal regions consistent with MN puncture remnants should be marked with arrows also in figures E-H. You can include this information at the end of the morphological description, or join them together to simplify the image description.
  • Results, page 15, line 562. “The skin incubated with calcein solution also exhibited”. This way you make connection with the previous HE images and you can avoid the sentence relating the findings with Figure 4A in lines 568-569, again to simplify. Please, check the commas in this paragraph, too.
  • Results, page 16, line 576-577. All MN skin samples appear to have invaginations on their surface (Figure….).
  • Results, page 16, line 577-578. “Green color represents the unspecific fluorescence caused by the calcein and can be related with the intensity of the calcein signal”.
  • Results, page 16, line 579. “no immunofluorescence was detected”.

Results, page 16, line 580-583. “The signal was morphologically less intense for 800 µm MNs  (Figures 5F and 5H) compared to 400 µm MNs (Figures 5E and 5G), suggesting a potential for systemic drug delivery, since…”. 

Round 3

Reviewer 3 Report

The article by Guimaraes et al. about the delivery of rivastigmine with microneedles was decisively improved and meritorious for publication in my opinion. However, there is still an issue:

  • When I pointed as minor issue there wasn´t a Figure S7, it was my intention to point that there actually was a Figure S7, but there wasn´t an histological picture. I was (and still am) pretty sure this supplementary file was confused. After reviewing the whole supplementary file and the section “supplementary materials” at the end of the main manuscript, in page 23, there should be 11 supplementary figures, but the provided supplementary file has only 10 figures. I checked three times that the supplementary file I have is the one provided by the editorial manager. Moreover, it is coupled to the version of the manuscript named “v3”. Please check again.
  • In addition, I would like to invite you to make subsections in the supplementary file for figures, tables… This way it would be easier to find the supplementary material.
  • Finally, a brief recommendation is that you can remove the corrections made in the previous versions of the review, leaving only the ones performed since the last revision. This way the text would be much easier to be read by the reviewers.

I am unable to detect other significant flaws in this revised version.
